# Structural basis for reactivating the mutant *TERT* promoter by cooperative binding of p52 and ETS1

Xueyong Xu[1], Yinghui Li[1], Sakshibeedu R. Bharath[1], Mert Burak Ozturk [1,2], Matthew W. Bowler [3,4], Bryan Zong Lin Loo[1], Vinay Tergaonkar[1,2,5] & Haiwei Song[1,2]

Transcriptional factors ETS1/2 and p52 synergize downstream of non-canonical NF-κB signaling to drive reactivation of the −146C>T mutant *TERT* promoter in multiple cancer types, but the mechanism underlying this cooperativity remains unknown. Here we report the crystal structure of a ternary p52/ETS1/−146C>T *TERT* promoter complex. While p52 needs to associate with consensus κB sites on the DNA to function during non-canonical NF-κB signaling, we show that p52 can activate the −146C>T *TERT* promoter without binding DNA. Instead, p52 interacts with ETS1 to form a heterotetramer, counteracting autoinhibition of ETS1. Analogous to observations with the GABPA/GABPB heterotetramer, the native flanking ETS motifs are required for sustained activation of the −146C>T *TERT* promoter by the p52/ETS1 heterotetramer. These observations provide a unifying mechanism for transcriptional activation by GABP and ETS1, and suggest that genome-wide targets of non-canonical NF-κB signaling are not limited to those driven by consensus κB sequences.

[1] Institute of Molecular and Cell Biology, 61 Biopolis Drive, Singapore 138673, Singapore. [2] Department of Biochemistry, National University of Singapore, 14 Science Drive, Singapore 117543, Singapore. [3] European Molecular Biology Laboratory, Grenoble Outstation, 71 Avenue des Martyrs, CS 90181, 38042 Grenoble, France. [4] Unit of Virus Host-Cell Interactions, Univ. Grenoble Alpes-EMBL-CNRS, 71 Avenue des Martyrs, CS 90181, 38042 Grenoble, France. [5] Centre for Cancer Biology, University of South Australia and SA Pathology, Adelaide 5001 SA, Australia. These authors contributed equally: Xueyong Xu, Yinghui Li. Correspondence and requests for materials should be addressed to V.T. (email: vinayt@imcb.a-star.edu.sg) or to H.S. (email: haiwei@imcb.a-star.edu.sg)

Telomeres are tandem TTAGGG sequence repeats found at chromosomal ends that are essential to preserve chromosomal integrity[1]. Telomeres are maintained by the telomerase ribonucleoprotein (RNP) complex[2]. In the absence of telomerase activity, gradual telomeric shortening eventually causes cellular senescence or cell death after progressive cycles of cell division[3]. The catalytic subunit, telomerase reverse transcriptase (TERT), and telomerase RNA component (TERC) constitute the core components of the telomerase ribonucleoprotein (RNP) complex that is essential for maintenance of telomere length[4]. While TERC is ubiquitously expressed in human and murine cells, TERT is transcriptionally silenced in somatic cells and is expressed in detectable amounts only in germ cells and stem cells[5]. Reactivation of TERT leads to cancer progression by canonical (telomere dependent) and non-canonical mechanisms[6].

As incessant synthesis of telomeric DNA is necessary for unlimited cancer cell proliferation, 85–90% of human tumors reactivate TERT expression[7]. Undoubtedly, understanding how the dormant TERT promoter is reactivated during carcinogenesis is central to understanding the mechanisms of cancer progression. Two highly recurrent, but mutually exclusive, cancer-specific TERT promoter mutations identified recently have been shown to cause reactivation of TERT promoter in cancers[8–12]. These two C>T mutations, located at either −146 or −124 base pairs (bp) upstream of the translational start site of TERT (referred to as −146C>T or −124C>T), lead to increased telomerase activity and now represent the most common noncoding mutations in cancer[13]. Clinically, these two mutations induce increased telomerase activity and decreased patient survival[14,15]. However, the mechanism by which these single-residue mutations in the human genome cause such a dramatic change in the fate of a cell, by reactivating the expression of the immortalizing enzyme, namely TERT, is poorly understood.

Both −146C>T and −124C>T mutations create an identical 11-bp sequence (5′-CCCCTTCCGGG-3′), containing a putative binding site for E26 (ETS family) transcription factors (GGAA, reverse complement)[9]. Subsequent studies have been performed to identify and characterize the transcription factors which are associated with the reactivation of these two mutant TERT promoters[16]. Recent data demonstrate that three ETS family members-GABPA, ETS1 and ETS2 can bind and activate the mutant TERT promoters[17–19]. The ETS family is a large family of transcription factors existing throughout the metazoan phyla. All ETS family members are defined by the highly conserved DNA-binding ETS domain, which binds similar DNA sequences with a central 5′-GGA(A/T)-3′ core (ETS motif)[20]. Given that there are 28 ETS family members in humans, the mechanism by which the mutant TERT promoters recruit specific ETS family members remains enigmatic. The specificity for GABPA may reside in its unique feature of obligate multimeric assembly. GABPA heterodimerizes with GABPB, two GABPA/GABPB heterodimers further dimerize through homodimerization of GABPB itself, resulting in a GABP heterotetramer capable of binding two ETS motifs[21]. On the mutant TERT promoters, one GABP heterodimer binds the mutant −146C>T or −124C>T ETS motif, the other binds a tandem flanking native ETS motif[17]. ETS1/2, unlike GABPA, only activate −146C>T mutant TERT promoter through cooperation with p52 downstream of non-canonical NF-κB signaling[18]. ETS2 is closely related to ETS1, and they have very similar domains and are essentially identical in their DNA-binding domains. ETS1/2 are in an autoinhibited state, brought about by the inhibitory module flanking the ETS domain[22]. DNA binding of ETS1/2 requires the relief of the autoinhibition by their partner transcription factors bound to adjacent sequences or by palindromic ETS sequence-mediated homodimerization[23–26].

p52 is a member of the NF-κB family of transcription factors which regulate gene expression in response to a wide array of signaling by binding to consensus κB sites of 5′-GGGRNYYYCC-3′ (R is a purine, Y is a pyrimidine, and N is any nucleotide)[27]. The −146C>T TERT promoter does not constitute a palindromic ETS sequence or a consensus κB site, nor does it contain any other known sequence bound by ETS1 partners. Indeed, how ETS1/2 are selectively recruited to the −146C>T TERT promoter and how they partner with p52 to activate the mutant TERT promoter is mechanistically unclear. To delineate this mechanism, here we report the crystal structure of p52/ETS1/−146C>T TERT promoter fragment complex. The structure offers a number of unexpected insights. Structural snapshots combined with cell-based functional assays provide insights into the mechanisms of −146C>T TERT promoter reactivation by p52 and ETS1/2. While understanding the reactivation of the mutant TERT promoter was the main aim of this study, our results also comprehensively document that NF-κB transcription factors can regulate transcription without specific DNA binding. These results therefore suggest that the number of NF-κB target genes, which are currently predicted based on existence of consensus κB sites in the upstream regions, could be significantly higher than currently estimated[27].

## Results

**Overall structure.** The major domains of ETS1 protein include the PNT domain, the acidic transactivation domain, the ETS DNA-binding domain and autoinhibitory module which consists of inhibitory helix 1 (HI1), inhibitory helix 2 (HI2), helix 4 (H4), and helix 5 (H5)[26]. p52 consists of the Rel homology domain and the glycine-rich region (Fig. 1a)[28]. To investigate the interaction between ETS1 and p52, we performed immunoprecipitation assays in HEK293T cells that were transfected with expression constructs for full-length and truncations of human p52 and ETS1. The region containing the ETS domain and inhibitory elements of ETS1 (residues 267–441), but not the region containing PNT domain and the acidic transactivation domain of ETS1 (residues 1–266), strongly binds to p52. The region of p52 (residues 1–350) containing RHD domain was sufficient to bind to ETS1 (Supplementary Fig. 1a, b).

We next determined the crystal structure of p52-RHD (residues 35–329) in complex with ETS1$_{331-441}$ and a 15 bp DNA fragment of TERT promoter containing −146C>T ETS motif (designated as p52/ETS1/−146C>T) (Supplementary Table 1). The final refined model contains the 15 bp DNA fragment, p52 residues 226–328 designated as the dimerization subdomain of p52-RHD (p52-DSD), and ETS1 residues 332–437 (Fig. 1b, c). The DNA-binding subdomain of p52-RHD (p52-DBSD, residues 35–225) is assumed to be disordered as its electron densities are not clear enough for unambiguous model building (Supplementary Fig. 2a). The disorder of p52-DBSD is not due to crystal packing as it can be roughly fitted into the fragmented electron densities without steric clashes with p52-DSD, ETS1, and DNA (Supplementary Fig. 2b). The ETS fragment consists of five α helices (H1, H2, H3, H4, and H5) and four-stranded antiparallel β sheets. H1, H2, and H3 and the β sheets comprise the ETS domain. The ETS domain is followed by the C-terminal inhibitory H4 and H5 α helices, which pack on the face of the ETS domain opposite the DNA-binding surface. H3 inserts into the major groove of the DNA and contacts the GGAA central core specifically. The p52-DSD is a seven-stranded β-barrel folding into immunoglobulin-like domain. Flanking both ends of the β-barrel, there are many loops linking antiparallel β-strands together. In the p52/ETS1/−146C>T ternary complex structure, ETS1 recognizes the −146C>T TERT promoter exclusively and concurrently interacts with p52-DSD.

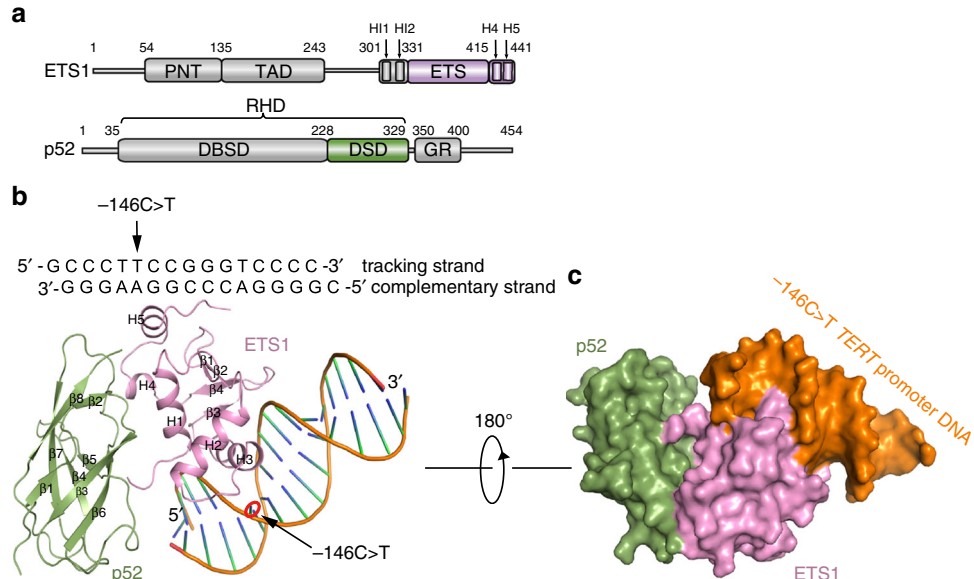

**Fig. 1** Overall structure of p52/ETS1/−146C>T mutant *TERT* promoter complex. **a** Color-coded domain architecture of human ETS1 and p52. The pink and green color schemes are used in the structural figures of p52/ ETS1/−146C>T *TERT* promoter fragment complex. PNT PNT domain, TAD acidic transactivation domain, ETS ETS DNA-binding domain, RHD Rel homology domain, DBSD DNA-binding subdomain of RHD, DSD dimerization subdomain of RHD, GR glycine-rich region. **b** Ribbon diagram of the ternary complex. **c** Surface representation of the ternary complex

**Interaction of p52 with ETS1**. The interaction of p52 and ETS1 buries a total solvent accessible surface area of 1187 Å$^2$. p52 uses the loops flanking one end of the β-barrel of p52-DSD to interact with α-helices H1, H4, H5, loop H4–H5 and N-terminal region of ETS1 domain predominantly through a network of hydrogen bonds and salt bridge contacts, supplemented with additional van der Waals contacts (Fig. 2a). The side chains of Arg241 and Asp275 of p52 form hydrogen bonds with backbone carbonyl group of Thr346 and backbone amide group of Ile335 of ETS1, respectively. Arg241 of p52 also forms salt bridges with Asp347 of ETS1. Met297 and Lys298 of p52 interact with Leu422 of ETS1, via backbone–backbone hydrogen bonds. These interactions are strengthened by a patch of van der Waals contacts between Leu342, Leu422, Tyr424, and Met432 in a hydrophobic cave of ETS1 and Met297 and Leu328 of p52. Met297 of p52 fits snugly into the hydrophobic cave. Although other hydrophobic residues (Leu421, Leu429, Leu433) of ETS1 in this hydrophobic cave do not directly form van der Waals contacts with p52, they help to stabilize the hydrophobic environment. To confirm these structural findings, we performed mutational analysis of p52. In the structure, the side chain of Arg241 forms salt bridge as well as hydrogen bond with ETS1. Consistent with its important role in the p52/ETS1 interaction, Flag pull-down assays with the purified proteins containing the same regions of p52 and ETS1 in the crystal structure showed that simultaneous mutation of Arg241, Met297, Lys298 participating in p52-ETS1 interaction to alanine (RMK>A) resulted in marked reduction of p52 affinity with ETS1 (Fig. 2b, Supplementary Fig. 3a). Additional mutation of Leu328 to Glu in the context of RMK>A (RMK>A L328E) further reduced the binding of p52 to ETS1 (Supplementary Fig. 3b). We also examined the binding thermodynamics of wild type (WT) and RMK>A p52 to ETS1 by isothermal titration calorimetry (ITC). WT p52 bound to ETS1 with an equilibrium dissociation constant ($K_d$) of 6.76 μM, whereas the triple mutant showed 5-fold reduced binding to ETS1 (Fig. 2c, d).

Sequence alignments indicate that the position of key residue Arg241 of p52 which is significant for the p52/ETS1 interaction is occupied by hydrophobic residues in the counterpart positions of p50, RelA and RelB. Furthermore, most of other residues of p52

in the p52/ETS1 interface are not conserved among NF-κB family members (Supplementary Fig. 4). We next checked the interaction between ETS1$_{331-441}$ and the RHD domains of other NF-κB family members (p50, RelA, RelB) by Flag pull-down assays. Although all the four NF-κB family members can interact with ETS1, p52 shows the strongest interaction with ETS1 (Supplementary Fig. 5). The strongest interaction between p52 and ETS1 may play an important role in the process whereby the p52 subunit downstream of non-canonical NF-κB signaling appears to be preferentially selected for synergizing with ETS1/2 to drive −146C>T mutant *TERT* promoter activation. However, the exact mechanism of this phenomenon remains to be further investigated. All the ETS domains of ETS family members are conserved. However, the inhibitory helices H4 and H5 of ETS1/2, which are important for p52 binding, are not conserved among the large ETS transcription factor family in humans (Supplementary Fig. 6), providing a plausible explanation of why only ETS1/2 are selected from ETS transcription factor family to function through the non-canonical NF-κB signaling to activate −146C>T *TERT* promoter. These results suggest a functional expansion of targets downstream of non-canonical NF-κB signaling and of targets regulated by ETS1/2 which have not yet been realized previously.

**ETS1 rather than p52 binds to −146C>T *TERT* promoter**. ETS1 specifically binds to −146C>T *TERT* promoter by inserting its H3 helix into the major groove of the ETS motif created by the mutation at −146 position much like that documented previously (Supplementary Fig. 7)[24–26]. p52 alone is unable to specifically bind the −146C>T *TERT* promoter as a consensus κB site which is necessary for specific DNA binding by NF-κB family members including p52 is absent from −146C>T *TERT* promoter. Consistent with this view, ITC data showed that in the absence of ETS1, p52-RHD exhibited nearly negligible binding to the 25 bp −146C>T *TERT* promoter DNA containing the potential p52-DBSD binding motif of 5′-TCCCC-3′, which is the part of the κB site (Supplementary Fig. 8a). In contrast, the specific binding affinity between NF-κB

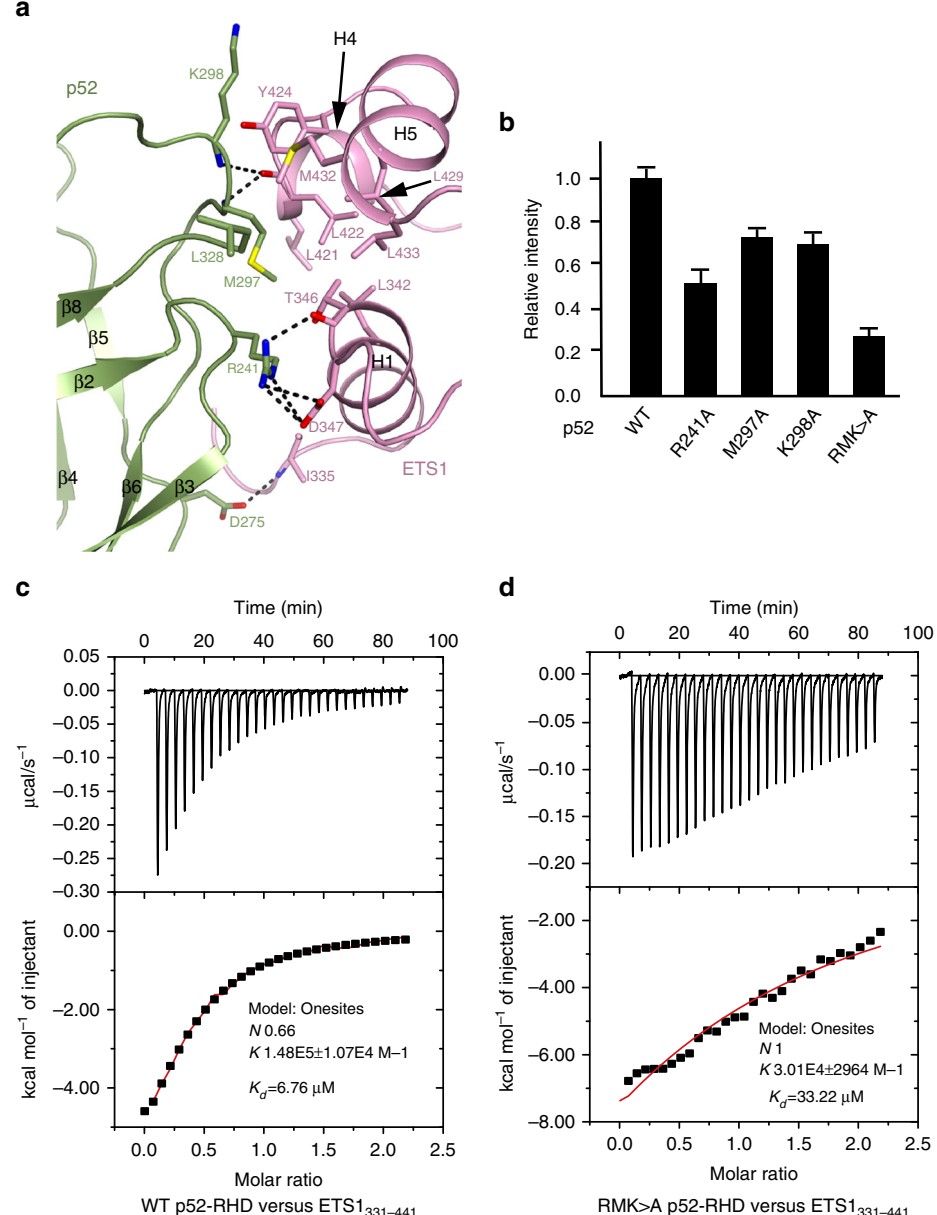

**Fig. 2** The p52/EST1 interface. **a** A close-up view of the atomic interactions between p52-DSD and ETS1$_{332-437}$. **b** FLAG pull-down assays of p52 and ETS1. The assays were quantified by band densitometry. Error bars, s.d. for triplicate experiments. **c**, **d** ITC titrations of ETS1$_{331-441}$ with p52-RHD and its RMK>A mutant. The upper panels show the binding isotherms and the lower panels show the integrated heat for each injection fitted to a single-site model

family members and the full κB sites are reported to be in the pM to nM range[29–31]. It has been proposed that in cooperation with ETS1, p52 may bind to −146C>T *TERT* promoter through specific interaction between p52-DBSD and the 5′-TCCCC-3′ site[18]. However, our structure shows the contact between p52-DBSD and the 5′-TCCC-3′ site is sterically blocked by p52-DSD and ETS1 (Fig. 3a). ITC data showed that p52-DBSD does not enhance the DNA binding of the ETS1/p52 complex (Fig. 3b, c). An ITC assay using a short 10 bp −146C>T *TERT* promoter containing only the ETS1-binding site showed that the flanking sequences of the ETS motif have no contribution to the DNA-binding affinity of the p52/ETS1 complex (Fig. 3b, Supplementary Fig. 8b). These observations suggest that p52 does not specifically bind to the −146C>T *TERT* promoter in vitro, regardless of whether ETS1 is present or not. These results provide structural evidence showing that p52 could activate transcription independent of DNA binding. Indeed,

experiments from the Perkins and Oren labs have hinted that p52 can activate transcription without DNA binding[32,33].

**p52 counteracts DNA binding autoinhibition of ETS1.** The DNA-binding affinity of ETS1 is regulated by its autoinhibitory module, which is formed predominantly by N-terminal inhibitory helices (HI-1, HI-2) and C-terminal (H4, H5) inhibitory helices flanking the winged helix–turn–helix ETS domain. Chemical shift data from NMR spectra shows that the HI-1 and HI-2 inhibitory helices pack against the H4 and H5 inhibitory helices and against helix H1 of the ETS domain[34]. This compact conformation is known to reduce the DNA-binding affinity of ETS1. However, this autoinhibited state is not immobile. Both hydrogen exchange measurements by NMR and proteolysis studies suggest that HI-1 and HI-2 inhibitory helices are only marginally stable and poised to unfold even in the absence of DNA[34]. The conformational fluctuation of HI-1 and HI-2 renders a chance to release HI-1 and

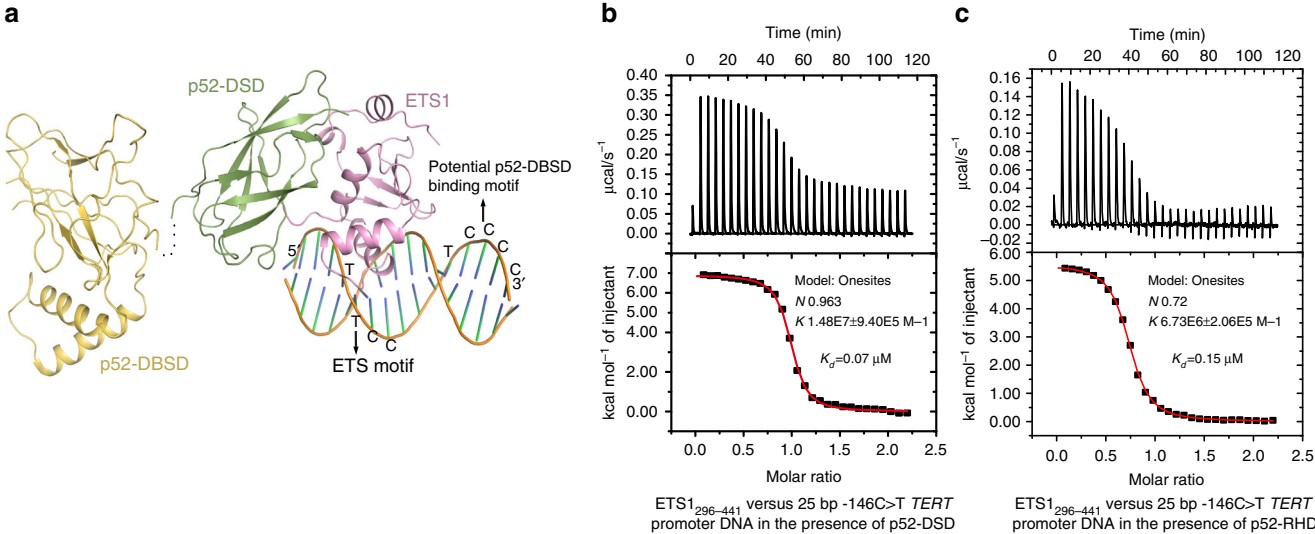

**Fig. 3** p52 does not specifically bind to −146C>T *TERT* promoter. **a** A model showing that p52 would not bind to −146C>T *TERT* promoter specifically. The relative orientation between p52-DSD and p52-DBSD showed here is the same as that in the structure of p52 homodimer in complex with DNA (PDB code: 1A3Q). **b, c** ITC titrations of ETS1$_{296-441}$ and a 25 bp −146C>T *TERT* promoter DNA in the presence of p52-DSD (**b**) or p52-RHD (**c**)

HI-2 inhibitory helices and to expose the face composed of H1, H4 and H5[34]. Then both of the released inhibitory helices and the exposed face can be contacted by another ETS1 molecule or by ETS1 partners to counteract the compact autoinhibited state of ETS1[24,26,35]. Structural superposition of ETS1 in the p52/ETS1/−146C>T complex and the autoinhibited ETS1 showed that p52-DSD and the HI-1 and HI-2 inhibitory helices pack against ETS1 residues 331–441 through overlapping surface regions, suggesting that the HI-1 and HI-2 inhibitory helices would interfere with the binding of ETS1 to p52-DSD (Fig. 4a). Consistent with this, our ITC data showed that ETS1$_{296-441}$ containing HI-1 and HI-2 showed threefold reduced binding to p52-RHD (Figs. 2c and 4b). On the other hand, binding of p52 to ETS1 would displace the HI-1 and HI-2 helices, therefore counteracting the DNA binding autoinhibition of ETS1. In line with this view, ITC data showed that in the presence of p52-DSD, ETS1$_{296-441}$ binds to −146C>T *TERT* promoter DNA with greater affinity. (Figs. 3b and 4c, Supplementary Fig. 8b, 8c). The mechanism by which p52 counteracts autoinhibition of ETS1 resembles that of Runx1[26].

**p52 and ETS1 can form a heterotetramer.** Homo and hetero-dimerization of NF-κB family members are necessary for their diverse physiological roles in vivo[27]. Within the crystal lattice of the p52/ETS1/−146C>T complex, two p52-DSD molecules homodimerize through a 2-fold crystallographic symmetry (Fig. 5a). Such a p52-DSD homodimer is essentially identical to that observed in the structure of p52 homodimer in complex with the κB site (Supplementary Fig. 9a)[28], suggesting that the p52/ETS1 heterodimer can dimerize further via p52 homodimeriza-tion to form a heterotetramer. Due to the weak interaction, p52 and ETS1 are always separated in two peaks in gel filtration (Supplementary Fig. 9b). In order to validate the presence of the p52/ETS1 heterotetramer in solution, we performed disulfide crosslinking and examined the molecular weight of the complex using blue native PAGE. The result showed that ETS1 and p52 can indeed form a heterotetramer in solution (Supplementary Fig. 9c, d). GABPA and GABPB, the obligate partners of GABP, also form a heterotetramer. In the GABP heterotetramer, the N-terminal ankyrin repeats of GABPB form a heterodimer with GABPA while the C-terminal leucine zipper-like domain forms a homodimer with itself. Between the N-terminal ankyrin repeats

and C-terminal leucine zipper-like domain, there is a long linker (Fig. 5b)[21]. In contrast, the p52/ETS1 heterotetramer is a rigid body whereby p52-DSD mediates both heterodimerization and homodimerization. Therefore, the GABP heterotetramer is much more flexible than the p52/ETS1 heterotetramer. The two GABPA/GABPB heterodimers can swing within a wide range of distance[17]. These findings have implications in the distinction between the activation of −146C>T and −124C>T mutant *TERT* promoters as discussed below.

**The role of flanking native ETS motifs in *TERT* reactivation.** Apart from binding to the mutation-generated ETS motif, GABP heterotetramer also interacts with one of the native ETS binding motifs flanking the mutation site (Fig. 5c)[17]. Since p52 and ETS1 can also form a heterotetramer that interacts with the −146C>T *TERT* promoter DNA containing two ETS binding motifs (Supplementary Fig. 9e), we next used −146C>T *TERT* promoter-driven luciferase reporters to investigate the role of native flanking ETS motifs in activation of −146C>T *TERT* promoter. To test whether the flanking ETS motifs are also required in addition to the p52/ETS1 bound to the mutant site for −146C>T *TERT* promoter activation, we mutated the flanking ETS motifs and examined their effects on −146C>T *TERT* promoter. All the three flanking ETS motifs were found to be necessary for full activity of −146C>T *TERT* promoter, although the effect of −190ETS was weaker (Fig. 5d). Since our results were suggestive of the fact that p52 could activate *TERT* transcription downstream of ETS and not κB sites, we next analyzed the impact of non-canonical NF-κB signaling in activating ETS1-dependent transcription of −146C>T *TERT* promoter. During activation of the non-canonical NF-κB pathway, stabilization of NF-κB-inducing kinase (NIK) triggers the processing of NF-κB p100 to p52, leading to the nuclear accumulation of p52. Overexpression of NIK alone results in a marked increase in −146C>T *TERT* promoter activity, which was further elevated with ETS1 overexpression, indicating that p52 and ETS1 can synergistically activate the −146C>T *TERT* promoter. All the three flanking ETS motifs were necessary for efficient activation of −146C>T *TERT* promoter by p52/ETS1 hetero-tetramer, as mutating any of the three ETS motifs reduced −146C>T *TERT* promoter activity significantly, either in the presence of stabilized NIK alone or during co-overexpression of

 

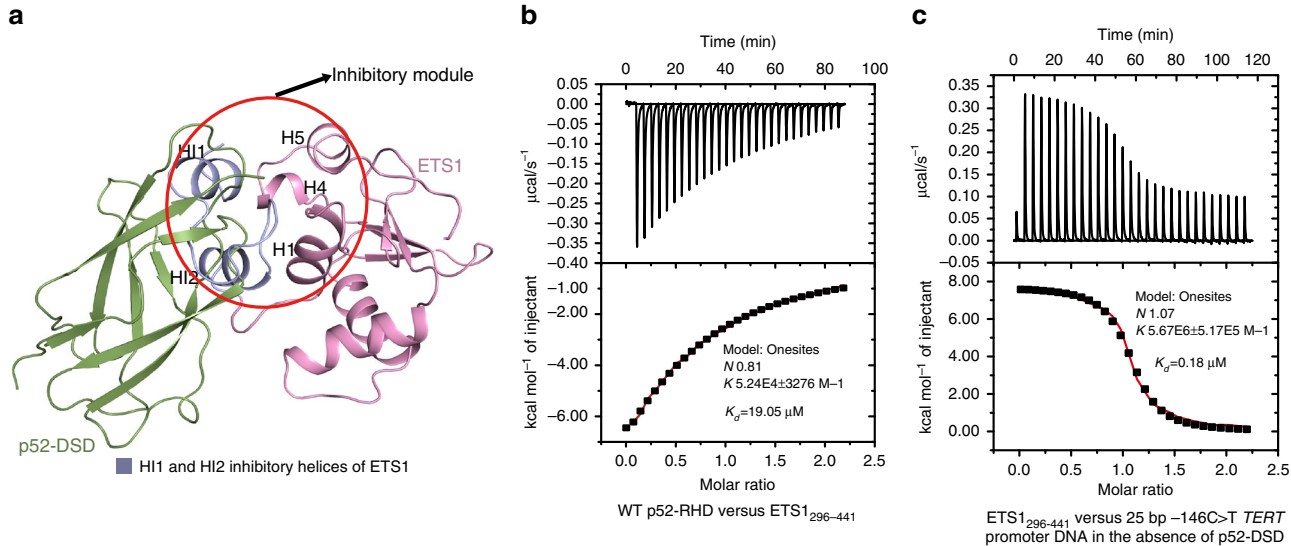

**Fig. 4** p52-DSD counteracts autoinhibited state of ETS1. **a** Superposition of the ETS1 molecule of the p52/ETS1/−146C>T complex onto the autoinhibited ETS1 (PDB code: 1R36) shows a steric clash between p52-DSD and the inhibitory helices HI1 and HI2 of ETS1. ETS1 in the p52/ETS1/−146C>T complex is omitted for clarity. **b** ITC titration of p52-RHD and ETS1$_{296-441}$ containing the HI1 and HI2 inhibitory helices. **c** ITC titration of ETS1$_{296-441}$ and a 25 bp −146C>T *TERT* promoter DNA in the absence of p52-DSD

NIK and ETS1 (Fig. 5e). Importantly, a p52 RMK>A mutant which markedly impaired the interaction with ETS1 cannot activate the −146C>T *TERT* promoter, suggesting that direct interaction between p52 and ETS1 is essential for activation of the −146C>T *TERT* promoter (Fig. 5f). In support of the crosstalk between p52 and ETS1 in mutant *TERT* promoter activation, p52 siRNA downregulation in NIK expressing cells led to a decline in ETS1 protein levels, along with the concomitant reduction of −146C>T *TERT* promoter activity (Fig. 5g). These observations were further substantiated by the significant impairment of −146C>T *TERT* promoter activity following ETS1 siRNA knockdown, highlighting the critical role of ETS1 in p52-mediated mutant *TERT* promoter activation (Fig. 5g). In contrast, the p52 RMK>A mutant activated a NF-κB2 reporter plasmid to a similar extent as wild-type p52 (Fig. 5h), illustrating that mutations in p52 RMK>A specifically disrupted the interaction of p52 with ETS1 that mediated ETS1-dependent transactivation of −146C>T *TERT* promoter. All these results reveal that the p52/ETS1 heterotetramer is capable of binding to two ETS motifs to activate −146C>T *TERT* promoter which could be activated downstream of non-canonical NF-κB signaling.

## Discussion

Three ETS family members, GABPA, ETS1/2 have currently been identified to activate the −124C>T and −146C>T mutant *TERT* promoter[17,18]. GABPA has been believed to be unique among the large ETS family, for only GABPA forms a heterotetramer with GABPB[20]. In this study, we show that p52 and ETS1 also form a heterotetramer. Analogous to the GABP heterotetramer, the p52/ETS1 heterotetramer can bind to two adjacent ETS motifs in the −146C>T *TERT* promoter. The −124C>T and −146C>T mutations occur at different frequency in cancers and the levels of *TERT* expression have been found to be different between the two mutations, suggesting that distinct mechanisms of *TERT* transcriptional activation exist at the two mutation sites. The −124C>T mutation is predominant in almost all cancers except skin cancers. In skin cancers, the −124C>T and −146C>T mutations occur at almost equal frequency[36,37]. These observations hint that different mechanisms occur in the regulation of

the two mutant *TERT* promoters for different cancers. It has been reported that the GABP heterotetramer activates both −124C>T and −146C>T *TERT* promoters, but the p52/ETS1 complex was found only to activate the −146C>T *TERT* promoter[17,18]. What causes this difference is not clear. Our structure shows that the p52/ETS1 heterotetramer is a rigid body, while GABP heterotetramer is much more flexible[21]. The assembly flexibility of GABP heterotetramer allows it to bind two adjacent EST motifs with various spacing[17]. The rigid p52/ETS1 heterotetramer, however, may bind two adjacent ETS motifs with strict spacing. In the −146C>T mutant *TERT* promoter, the spacing between −146C>T ETS motif and the flanking native motifs is around 50 base pairs. It is possible that −124C>T is too close to the flanking −91ETS and −96ETS native motifs that p52/ETS1 heterotetramer cannot bind to two ETS motifs within such a short distance.

It is widely accepted that NF-κB members specifically bind to consensus κB sites to activate transcription[27]. However, there are no consensus κB sites in the WT or mutant *TERT* promoter for binding by p52. During p52/ETS1-dependent reactivation of *TERT* at −146C>T *TERT* promoter, p52 does not contact DNA to function as a transcription factor. Lacking an activation domain, p52 homodimers alone cannot activate transcription. It functions as an ETS1 partner to counteract ETS1 autoinhibition and as an adaptor to bridge ETS1 to form p52/ETS1 heterotetramer. This atypical function of p52 suggests that non-canonical NF-κB signaling is not limited to regulating the genes harboring consensus κB sites.

Given that both ETS and NF-κB family are subjects of intense investigation over two decades, it is surprising that the interaction between p52 and ETS1 has not been characterized before. Our results suggest that their interaction is weak[38], giving a possible explanation why it has been a challenge to characterize. However, this interaction could be specifically required to activate genes with special promoters like *TERT* which are activated by long-range interactions of multimeric transcription factors[39]. In addition, the weak nature of this interaction could be required to only mildly activate *TERT* because only 1000 molecules of *TERT* are present in a cell and that hyperactivation of *TERT* by stronger activators could cause adverse effects in cancer cells[40,41].

Furthermore, since both ETS1 and p52 play key roles in many physiological processes, the weak interaction which can be easily formed and dissociated could provide large ETS1 and p52 reservoirs which are readily available to bind to a multitude of their canonical partners. ETS1 binds to similar sequences with a central 5′-GGA(A/T)-3′ core. As this kind of sequences spread widely in the human genome, the function of p52/ETS1 heterotetramer may not be confined to reactivating the −146C>T *TERT* promoter. It is probable that the p52/ETS1 heterotetramer also binds to other sequences in the genome harboring two adjacent ETS motifs with appropriate spacing. This suggests that non-canonical NF-κB signaling and ETS1/2 may cooperatively

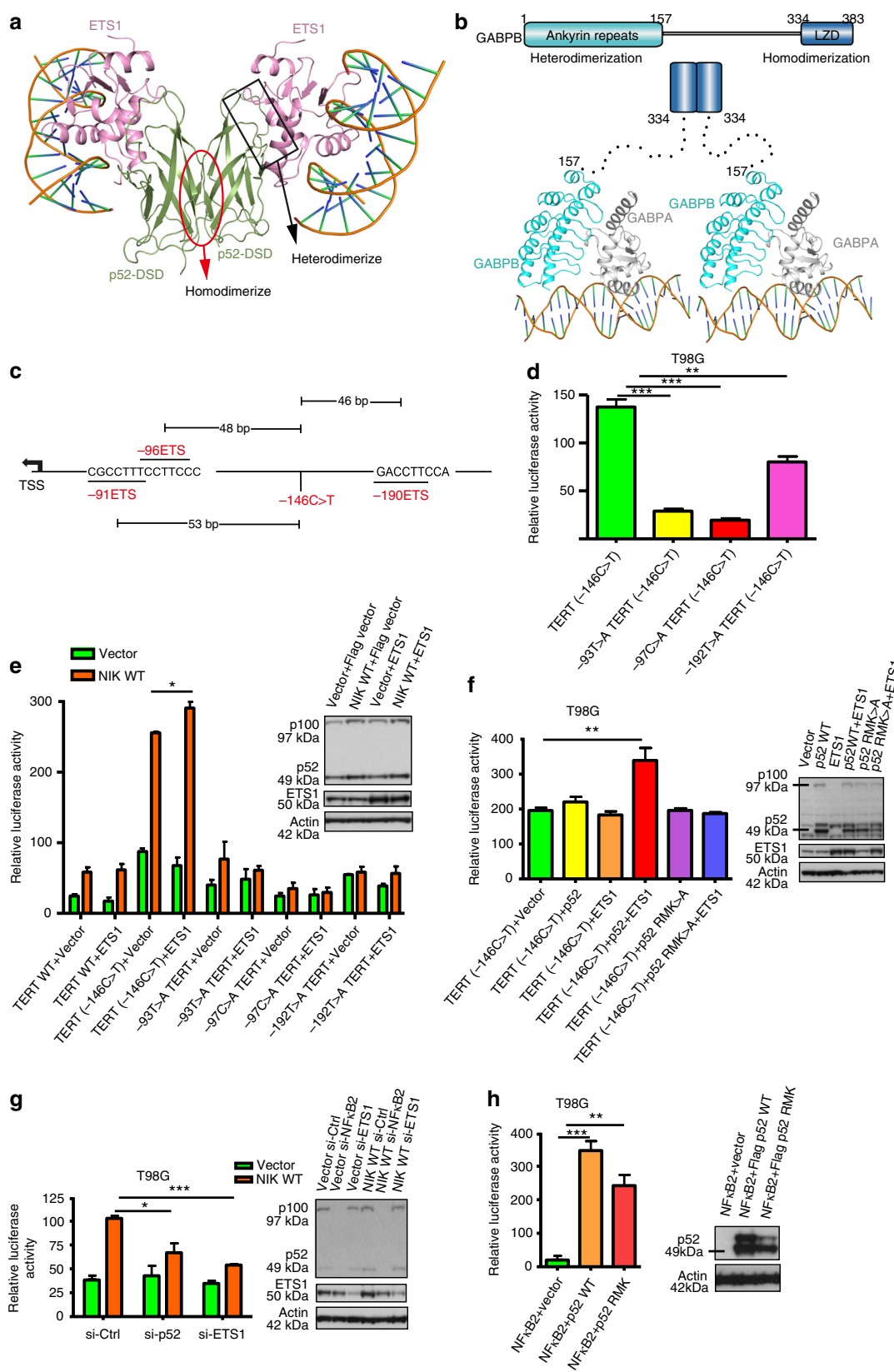

regulate the expression of many other genes besides *TERT*. Just as their similar function in activation of mutant *TERT* promoter, p52/ETS1 heterotetramer and GABP heterotetramer may carry out overlapping functions in regulating other genes.

## Methods

**Plasmids and DNA preparations**. All the constructs for expression in *Escherichia coli*, which include the RHD domains of RelA (residues 16–291), RelB (residues 111–405), p50 (residues 1–365), and p52 (residues 35–329) as well as p52-DSD (residues 226–328) and ETS1$_{296–441}$ and ETS1$_{331–441}$, were cloned into the vector pGEX-6P-1 (GE Healthcare). For Co-immunoprecipitation assay, full-length p52, full-length ETS1, p52$_{1–350}$, ETS1$_{1–266}$ and ETS1$_{267–441}$ were cloned into the pBobi mammalian expression vector with either HA or Flag tag at the N-terminus. Luciferase reporter constructs for wild-type *TERT* promoter and −146C>T *TERT* promoter were cloned into the pGL3 basic vector as reported previously[15]. Site-directed mutagenesis of p52 in pBobi vector and −146C>T *TERT* promoter reporter constructs in pGL3 vector were performed using the QuikChange Lightning Multi Site-directed Mutagenesis Kit (Agilent Technologies). The NF- κB2 luciferase reporter construct was generated by cloning the NF-κB2 promoter region (−1869 to −1302) into the pGL3 basic vector via XhoI and HindIII restriction enzyme sites.

All of the DNA duplexes used in this study were annealed in a buffer containing 10 mM Tris-HCl pH 7.5 and 10 mM DTT at concentration of 100 μM. Annealing was performed by heating at 95 °C for 10 min and then put on ice for 10 min. The sequences of the DNA duplexes and mutagenesis primers used are listed in Supplementary Table 2.

**Protein purification**. All the constructs are expressed in *Escherichia coli* strain Rosetta2 (DE3) by induction with 0.3 mM isopropyl β-D-thiogalactopyranoside (IPTG) at 18 °C overnight. The proteins were initially purified using GST affinity chromatography, followed by PreScission protease cleavage at 4 °C overnight. The proteins were further purified using heparin affinity chromatography and size exclusion chromatography. To assemble the complex, p52-RHD, ETS1$_{331–441}$ and the 15 bp DNA were mixed in a 1:1:1 molar ratio, followed by dialyzing overnight against a buffer containing 10 mM Tris-HCl, pH 7.5 and 10 mM DTT. The sample was then concentrated to 15 mg ml$^{−1}$ for crystallization.

**Crystallization and structure determination**. Crystals of the p52/ETS1/−146C>T complex were grown at 18 °C using the setting drop vapor diffusion method by mixing equal volumes of the complex and reservoir solution containing 100 mM HEPES, pH 7.0, 2.0 M ammonium sulfate. Crystals were soaked in a cryoprotectant buffer containing mother liquor and additional 25% glycerol (v/v) and were flash frozen in liquid nitrogen. X-ray diffraction data were collected at beamline PX-I (SLS, PSI, Switzerland) and by the autonomous ESRF beamline MASSIF-1 using automatic protocols for the location and optimal centering of crystals (ESRF, Grenoble, France)[42,43]. The beam diameter was selected automatically to match the crystal volume of highest homogeneous quality, in this case 30 μm. Strategy calculations accounted for flux and crystal volume in the parameter prediction for complete data sets[44]. The data were processed using XDS[45] and Aimless[46]. The initial phases were obtained by molecular replacement using PHASER[47] with search model prepared from PDB ID 1K78. The model of the complex was built using Coot[48], and then refined by Phenix.refine[49] and Refmac[50]. A representative portion of electron density map is shown in Supplementary Fig. 10.

**Isothermal titration calorimetry**. ITC measurements were performed at 8 °C using MicroCal VP-ITC (MicroCal Inc.). All protein and DNA samples were dialyzed into a buffer containing 25 mM Tris-HCl pH 7.5, 200 mM NaCl and 3 mM TCEP. To obtain the binding affinity between p52 and ETS1, 20 μM ETS1$_{296–441}$ or ETS1$_{331–441}$ (in cell) was titrated with 200 μM p52-RHD (in

syringe). To obtain the binding affinity between ETS1$_{296–441}$ with 10 bp or 25 bp DNA in the absence or presence of p52-DSD, 10 μM 10 bp or 25 bp DNA in the absence or presence of 100 μM p52-DSD (in cell) was titrated with 100 μM ETS1$_{296–441}$ (in syringe). To obtain the binding affinity between p52-RHD and 25 bp DNA, 10 μM 25 bp DNA (in cell) was titrated with 100 μM p52-RHD (in syringe). To obtain the binding affinity between ETS1$_{296–441}$ and 25 bp DNA in the presence of p52-RHD, 10 μM 25 bp DNA in the presence of 100 μM p52-RHD was titrated with 100 μM ETS1$_{296–441}$(in syringe). The titration comprised 29 injections of 10 μl each, separated by 240 s equilibration time. The datasets were analyzed using the Origin 7.0 program, fitted to a single-site binding model.

**Co-immunoprecipitation**. 293T HEK (ATCC® CRL-3216™) cells were seeded at a density of $8 \times 10^5$ cells per dish in 6 cm culture dishes and transfected the following day using X-tremeGENE HP DNA transfection reagent (Roche). Cells were co-transfected with 1.5 μg each of Flag- or HA-tagged p52 and ETS1 constructs (full length, deletion mutants or amino-acid substitution mutants) and harvested for co-immunoprecipitation assays after 3 days. Protein lysis buffer (10 mM Tris at pH 8, 170 mM NaCl and 0.5% NP40, supplemented with protease inhibitors) was added to cell pellets, incubated on ice for 15 min prior to sonication for 5 cycles. Lysates were then centrifuged at 14,000 rpm for 15 min before aspirating 1 ml of each supernatant for immunoprecipitation. Immunoprecipitation of Flag-tagged proteins was performed using 20 μl of anti-Flag M2 magnetic beads (packed gel volume) (Sigma: M8823) per sample. Eluted and input protein samples were loaded on a 4–12% Bis-Tris gel (Invitrogen) and immunoblotting was performed using the following antibodies diluted by 1000 times: anti-Flag (Sigma: F7425), anti-HA (Covance: MMS-101P), anti-Actin (Santa Cruz: sc-1616), anti-HSP90 (Santa Cruz: sc-13119). Uncropped scans of all blots are shown in Supplementary Fig. 11.

**Flag pull-down assay**. Flag-tagged ETS1$_{331–441}$ was incubated with p52-RHD and its mutants, RelA-RHD, RelB-RHD, and p50-RHD in a binding buffer consisting of 50 mM Tris HCl pH 7.4 and 150 mM NaCl for 2 h at 4 °C. The protein samples were then immobilized on 20 μl Anti-Flag M2 magnetic beads (Sigma: M8823) for 1 h at 4 °C. The beads were washed thrice with the binding buffer. The bound proteins were eluted with 250 μg ml$^{−1}$ FLAG peptide and subjected to SDS–PAGE and stained with Coomassie brilliant blue.

**Disulfide crosslinking and blue native PAGE**. Glu343 of ETS1$_{331–441}$ and Arg241 of p52-RHD whose Cα-Cα distance is in the range of that of disulfide bond were mutated to cysteine. At the same time, cysteine on the surface of p52-RHD and ETS1$_{331–441}$ were mutated to serine to prevent non-specific crosslinking. The resultant mutant proteins, p52-RHD (R241C, C57S, and C83S) and ETS1$_{331–441}$ (E343C and C350S) were kept in a buffer containing 10 mM DTT and then mixed at a molar ratio of 1:2, followed by desalting to remove DTT. The disulfide crosslinking of these two proteins was initiated by addition of 30 μM CuCl$_2$ and 100 μM phenanthroline for 1 h at 25 °C. The crosslinking reaction was stopped by the addition of 300 μM EDTA for 15 min at 25 °C. Reversal of crosslinking was carried out by adding 20 mM DTT for 1 h at 25 °C. The crosslinked and reversed products were checked by 4–16% gradient blue native PAGE[51].

**Electrophoretic mobility shift assay**. Crosslinked p52/ETS1 heterotetramer band was excised from blue native PAGE and mashed by squeezing with a syringe. The gel debris was soaked in double the volume of elution buffer (25 mM Tris pH 7.5 and 150 mM NaCl) overnight. The supernatant containing p52/ETS1 hetero-tetramer was incubated with FAM labeled 60 bp long −146C>T *TERT* promoter DNA for 1 h at 4 °C in a buffer containing 20 mM Tris HCl pH 7.5 and 150 mM NaCl. The samples were subjected to 4–16% gradient native PAGE and detected by fluorescein signal.

**Luciferase reporter assay**. T98G human glioblastoma cells (ATCC® CRL-1690™) were seeded at a density of $6 \times 10^4$ cells per well in 12-well plates and transfected

**Fig. 5** Binding of p52/ETS1 heterotetramer to the −146C>T mutant *TERT* promoter. **a** A ribbon diagram showing a p52/ETS1 heterotetramer bound to −146C>T *TERT* promoter DNA observed in the crystal lattice. The regions of p52 that heterodimerize with ETS1 and homodimerize with itself are indicated. **b** Color-coded domain architecture of mouse GABPB and a model of GABP heterotetramer (PDB code of the crystal structure showed here: 1AWC). LZD leucine zipper-like domain. **c** −146C>T mutation-generated and the tandem flanking native ETS motifs in the core *TERT* promoter. **d** The effect of mutation of adjacent ETS motifs (−93T>A, −97C>A, −192T>A) on the activation of mutant *TERT* promoter. Western blot analyses were performed in parallel to assess the levels of p100/p52 and ETS1 in luciferase reporter assay cells, with actin as a loading control. **e** The effect of mutation of adjacent ETS motifs (−93T>A, −97C>A, −192T>A) on the ETS1 and p52-mediated activation of mutant *TERT* promoter. **f** The effect of RMK>A mutant p52 on the activation of mutant *TERT* promoter. **g** The effect of p52 or ETS1 siRNA during non-canonical NF-κB signaling in regulation of the −146C>T mutant *TERT* promoter. **h** The effect of RMK>A mutant p52 on the activation of NF-κB2 reporter plasmid. *$P < 0.05$; **$P < 0.01$; ***$P < 0.001$; Student's *t*-test, two-tailed. Error bars in all luciferase reporter assays refer to standard deviations (s.d.) obtained from three independent experiments

the following day using X-tremeGENE HP DNA transfection reagent (Roche). For reporter assays with NIK, ETS1, p52 WT, or p52 RMK>A expression, T98G cells were infected with the respective lentiviruses to achieve stable expression of the constructs 3 days prior to seeding in 12-well plates. For siRNA treatment, T98G cells were transfected with RNA-lipid complexes of non-targeting control siRNA, NF-κB2 siRNA (L-003918) or ETS1 siRNA (L-003887) (Dharmacon smartpool siRNA) using X-tremeGENE siRNA transfection reagent (Roche) 24 h prior to transfection with luciferase reporter plasmids. Cells were transfected with 0.3 μg each of pGL3 WT *TERT* promoter or pGL3 −146C>T *TERT* promoter luciferase reporters and 25 ng of Renilla plasmid (Promega) per well. Cells were assayed for luciferase activity 3 days post transfection using the dual luciferase reporter kit (Promega). Triplicate wells were measured for each sample and relative luciferase activity was quantified by normalization of firefly luciferase activity to renilla luciferase activity or protein concentration. Western blot analysis of NIK, ETS1, p52 WT, or p52 RMK>A expressing cells was performed using the following antibodies diluted by 1000 times: anti-NF-κB p100/p52 (Cell Signaling; 4882), anti-ETS1 (Santa Cruz; sc-350), anti-Flag (Sigma; F7425), and anti-Actin (Santa Cruz; sc-1616).

**Data availability**. The atomic coordinates and structural factors for p52/ETS1/ −146C>T complex have been deposited with the Protein Data Bank under accession code 5ZMC, respectively. All other data supporting the findings of this study are available from the corresponding authors on reasonable request.

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

## Acknowledgements

We would like to thank the beamline scientists at MASSIF-1 (ID30A-1) of ESRF, France for assistance and access to synchrotron radiation facilities. This work was financially supported by the Agency for Science, Technology and Research and Singapore National Research Foundation under its Competitive Research Programme (NRF-CRP17-2017-02) in Singapore.

## Author contributions

H.S. and V.T. conceived and coordinated the study. X.X., Y.L., S.R.B., M.B.O, M.W.B, and B.Z.L.L. performed the experiments. X.X., Y.L., V.T., and H.S. analyzed the data and wrote the manuscript.

## Additional information

**Competing interests:** The authors declare no competing interests.

