## [Peer Review File · Nature Communications]

Reviewers' comments:

Reviewer #1 (Remarks to the Author):

This is a very interesting manuscript that describes the structure of the p52 NF- κ B subunit/Ets1 complex bound to a mutant form of the Tert promoter found in cancer. As the authors correctly identify, although there have been reports previously of p52 and other NF- κ B subunits being recruited to promoters and regulating transcription independently of DNA binding, this is the first report to my knowledge of the structure of such a complex being defined. There are numerous implications that arise from this paper for how NF- κ B dependent transcription can occur independently of canonical NF- κ B binding sites as well as the mechanistic basis of crosstalk between NF- κ B and Ets1.

Overall I found the data in this paper to be strong and convincing. There are, however, some areas where the manuscript requires strengthening to support some of the authors' conclusions.

Major comments

(1) In Figure 2B, data is presented showing the effect of p52 mutants on the interaction with Ets1. It is concluded that the R241A mutation strongly reduces this interaction. Figure 2B shows the quantification of data from IPs and an example gel is shown in Supp Fig 3. However, in this latter figure it can be seen that the level of input proteins (both Ets1 and p52) is significantly lower for the R241A assay. Although the quantification in Fig 2B takes this into account this does mean that at face value the R241A experiment is not comparable to the others as the concentration of protein in the IP will be lower. Was this lower concentration of input proteins for the R241A consistent across different repeats? When the authors refer to data being from 3 repeats, were these biological repeats where extracts were prepared and used from separate cell transfections? Or were they technical replicates of separate IPs from the same protein extract? If the latter, then 3 biological repeats need to be performed to generate this data. If the former, is there an example of a co-IP where the levels of p52 and Ets1 in the R241A samples were comparable to the other mutants being tested?

(2) In Figure 3 the authors conclude that p52 is not contacting a putative p52 binding motif present in the Tert promoter DNA sequence used. I agree with this conclusion. However, the DNA sequence used does not extend in the other direction to where p52 is found in the structure. It is possible therefore that p52 makes contacts with the DNA in this region of the promoter (even if it does not appear as a putative κ B site). It is not clear to me if this sequence was present in the 25 bp Tert promoter oligo used in the ITC experiments in Fig 2C & D. It would, however, be informative if these experiments could be repeated with oligos where the sequences 5' and 3' to the Ets1 binding site are removed to clearly determine if there is a contribution of flanking sequences to the DNA binding affinity of the p52/Ets1 complex.

(3) Figures 5D-F present luciferase assay data analysing the functional effects of p52 and Ets1 on Tert

promoter activity. There are a number of concerns about this data.

(a) The figure legend states that these are the average of two independent experiments. This is insufficient and further biological replicates should be performed so that statistical analysis of the data can be performed

(b) In Fig 5E and 5F, the additional effects of either expressing NIK or p52 respectively are quite small. This may result from endogenous levels of p52 in the cells, which the authors could assess by western blot combined with p52 siRNA to assess the extent to which the effects of Ets1 are dependent on any endogenous non-canonical pathway signalling. Nonetheless this data would be strengthened if the authors could examine endogenous levels of Tert.

(c) The strong effect of the RMK mutant in Fig. 5F supports the idea above that endogenous p52 is active in these cells. However, this experiment lacks a western blot control to demonstrate that the p52 RMK mutant is not affecting the levels of transfected Ets1 in these assays.

(d) A control missing from these experiments is the effect of the p52 RMK mutant on activation of an NF-kB reporter plasmid. Do these mutations specifically affect Ets1 dependent transactivation of the mutant Tert promoter or do they have more general effects on p52 dependent gene expression.

(4) An important experiment missing from this study is ChIP analysis to demonstrate p52 recruitment to the endogenous mutant Tert promoter in cells and whether this is affected by the RMK mutant. as would be predicted from the authors' data.

Minor concerns

(5) In Fig 2B it should be clarified if the full length p52 and Ets1 proteins are being expressed in this assay

(6) In Fig 2C the amino acids of the p52 RHD protein should be specified (as different ones are used in the manuscript)

(7) In the last paragraph of the introduction the authors state that this manuscript provides evidence for the first time that NF-kB transcription factors can regulate gene expression without binding DNA. However, as discussed by the authors later, this is not the first evidence. However, it is the first structural proof of this.

(8) In the discussion in the text of Fig, 5E the authors refer to p52 and Ets1 synergistically activating transcription from the Tert promoter. But as discussed above, the effects of p52 are quite small in these assays and certainly not synergistic.

(9) The manuscript lacks page numbers.

Reviewer #2 (Remarks to the Author):

The manuscript “Structural basis for reactivation of -146C>T mutant TERT promoter by cooperative binding of p52 and ETS1/2” is based on novel structural studies that show interaction of ETS1/2 transcription factors at the de novo site created by a C>T single nucleotide mutation at -146 bp position from the ATG start site. Previously, the same group had shown that non-canonical NF-kB signaling and ETS1/2 cooperatively drive TERT promoter activation with C>T mutation at the -146 bp position from the ATG site.

In this study, experimental evidence has been provided to show that p52 instead of binding to DNA acts as a co-regulator of ETS1 to inhibit auto-inhibition of ETS1 and forms p52/ETS1 heterotetramer that requires native flanking ETS motifs for sustained activation of the mutant promoter with C>T mutation at the -146 bp position. The impact of this novel observation can be more general and beyond TERT activation as this is the first study showing the inhibitory properties of p52 to counteract auto-inhibition of ETS1 with relevance for activation of promoters that lack kB consensus sites. Interaction between p52 and ETS1 has not been reported previously and as stated in the manuscript could be due to the weak interaction. Authors have also explained the probable reason for interaction of p52-ETS1 heterotetramer specifically with the mutant sequence at -146 bp and not at -124 bp position.

Ever since the discovery of TERT promoter mutations, the observational data have been consistent. The C>T mutations at -124 and -146 bp, mutually exclusive, occur at high frequencies in cancers that arise from tissues with low rates of self-renewal that include melanoma, bladder cancer, glioma, hepatocellular carcinoma and others. In all cancers with the exception of skin, the -124C>T mutation is predominant. In skin cancers, melanoma and non-melanoma, the two mutations -124C>T and -146C>T occur at almost equal frequencies with -146C>T in some studies shown to occur at slightly higher frequency than the -124C>T mutation. While tumors with either of the two TERT promoter mutations have consistently been shown to have increased TERT expression compared to tumors with those mutations, one study on glioma clearly showed that the increase in TERT expression due to -124C>T mutation was statistically significantly higher than the -146C>T mutations. Those observations point to different operative mechanisms for the two TERT promoter mutations despite creation of similar binding motifs for ETS transcription. It is important authors mention these facts in the manuscript in support of their results either in the introduction or in the discussion section.

The manuscript per se in the current form is quite diffuse, requires streamlining and should include the observations made in the previous paragraph.

Authors should provide values for statistical significance for the results from reporter assays and state whether differences in promoter activities for different constructs were statistically significant.

Reviewer #3 (Remarks to the Author):

In this paper, Xu et al. describe biochemical and structural studies of a TERT promoter DNA-bound ETS1-p52 transcription factor (TF) complex. This promoter is activated during cancerogenesis by two mutations that create binding sites for the ETS1-p52 TF complex and the authors aim to understand how this complex selectively recognizes the mutant promoter TF binding site and thus contributes to upregulation of TERT expression in cancer.

It has long been known that members of the ETS and NFkB family of TFs engage through evolutionarily conserved physical interactions to regulate inducible gene expression. How such these TFs cooperate to read out composite DNA binding sites is an important problem in the combinatorial control of gene expression. The authors show convincingly that the dimerization domain of p52 interacts with the DNA binding domain (DBD) of ETS1. Surprisingly, only the ETS1 DBD but not that p52, contributes specific DNA contacts on the co-crystallized TERT promoter fragment. The authors perform biochemical experiments and reporter gene assays to confirm the relevance of the interaction for transcriptional regulation of the TERT promoter. These data indicate that p52 prevents ETS1 autoinhibition and allows tetramerization of the complex through a p52-DD homodimerization interface without itself contributing to DNA binding. These data thus indicate that members of the NFkB family of TFs can contribute to the regulation of gene expression even while not engaging in direct DNA contacts.

It is of no doubt that this work will be of considerable interest to specialists in ETS and NFkB signaling. There are however reservations that arise from the fact that only limited fragments of ETS1, p52 and the DNA elements are co-crystallized and that the biochemical analysis is incomplete, as indicated below. While the fragments are sufficient for the interaction between ETS1- p52 and ETS1-DNA, it remains unclear whether a more extensive cooperative interface of the dimer is formed on an extended single-site DNA substrate, or of the tetramer on an extended two-site DNA substrate. The conclusion that the p52-DBD does not contribute to DNA binding seems premature and requires further analysis that should be readily doable with the available material in the authors' laboratory. Thus the study is promising and I would encourage resubmission of a revised manuscript. Upon revision, the authors should tighten up the manuscript, in particular the Discussion section. Nature Comm should do a careful copyediting job if the MS is accepted, as there are grammatical errors that impede effective communication.

Major comments:

1. The mutagenesis results shown in Fig. 3 only partially explain binding selectivity: Why does the triple mutant (RMK>A) of p52 show residual binding to ETS1? Would this indicate that the interface is more 'pleiotropic' than assumed by the authors and that other NFkB family members (p50, RelA, RelB) interact with ETS1? Can they test this possibility? Bacterial expression constructs for the RHD domains are readily available in the field.
2. Fig. 3: The authors suggest that the p52-DBSD does not engage the DNA binding site because the potential binding is sterically blocked. Could this be due to crystal packing and the preferential orientation of the co-crystallized complex on the 15bp DNA duplex that enabled successful

crystallization? While the authors show DNA binding data that demonstrate that the isolated p52-RHD does not support DNA binding on the minimal 25bp DNA substrate, it remains unclear if the p52-RHD contributes to DNA binding through a cooperative effect involving ETS1. The authors need to show DNA affinity measurements of ETS1 in the presence of p52-RHD to support the conclusion that the p52-DBSD does not contribute (even non-specifically) to DNA binding avidity. This can be easily done by a repeat experiment similar to that shown in Fig. 4C, but using the entire p52-RHD. If there is indeed no contribution to DNA binding by the p52 DBD in solution in the presence of ETS1, the physiological relevance of the unusual arrangement in the crystal structure would be more believable.

3. p.10, bottom paragraph: It is not clear how counteraction of ETS1 autoinhibition by p52 enables stronger binding of ETS1 to DNA...! Please clarify.
4. The authors need to confirm that the p52-ETS1 complex forms a heterotetramer in solution eg. by performing gel filtration/SEC-MALS etc.
5. Fig. 5a: The DNA configuration in the proposed tetramer is unusual and very different from what is proposed for (the more reasonable) arrangement of the GABPB-GABPA tetramer shown in Fig. 5b. DNA is one of the stiffest of known polymers with a persistence length of ~150 bp. The required DNA contortions would be expected to be energetically unfavorable and it is not clear how the proposed arrangement would enable a ETS1-p52 tetramer to bind two ETS motifs. This casts further doubt on the physiological relevance of the proposed tetramer.
6. Considering that the two ETS sites are within reasonably close proximity (~50-60bp), it would be important to test binding of the ETS1-p52 tetramer to such two-site substrates, eg. by EMSA analysis. If the tetramer cannot bind to such sites, as discussed on p.14, is it the TERT promoter even the physiologically relevant binding site?
7. It is not clear to this reviewer that the authors have sufficiently rigorously demonstrated that p52 in the p52-ETS1 complex does not contact DNA.
8. For ITC data analysis: What model was used for fitting of experimental data? What are the error values and binding stoichiometry?

Minor comments:

1. Please add page and line numbers to MS.
2. p.2, Line 1: Reference is missing supporting the statement that these are the 'most common genetic alterations in cancer'
3. p.2, Line 4: The mutations do not reactivate the enzyme TERT but 'expression of' the enzyme. Please correct.
4. p.2, Second paragraph: Reference is missing supporting the statement:...' reactivation of these two mutant promoters'.
5. p.5, Second paragraph: Replace '44' with '441'
6. p.5, Second paragraph and Fig.1: Please use consistent naming of subdomains in text and figures: Eg. p52-DBSD (text) and DBD (Figure) for the same segment.
7. p.9, bottom paragraph:...conformation will 'inhibit DNA binding' or 'reduce DNA binding affinity' but not 'inhibit DNA binding affinity'!
8. Table 1: Refinement section: Rfree value should not be in parenthesis as this indicates the 'values in the highest resolution shell'

Reviewer #1 (Remarks to the Author):

This is a very interesting manuscript that describes the structure of the p52 NF-kB subunit/Ets1 complex bound to a mutant form of the Tert promoter found in cancer. As the authors correctly identify, although there have been reports previously of p52 and other NF-kB subunits being recruited to promoters and regulating transcription independently of DNA binding, this is the first report to my knowledge of the structure of such a complex being defined. There are numerous implications that arise from this paper for how NF-kB dependent transcription can occur independently of canonical NF-kB binding sites as well as the mechanistic basis of crosstalk between NF-kB and Ets1.

Overall I found the data in this paper to be strong and convincing. There are, however, some areas where the manuscript requires strengthening to support some of the authors' conclusions.

We sincerely thank the reviewer for the complimentary remarks and positive appraisal.

Major comments

(1) In Figure 2B, data is presented showing the effect of p52 mutants on the interaction with Ets1. It is concluded that the R241A mutation strongly reduces this interaction. Figure 2B shows the quantification of data from IPs and an example gel is shown in Supp Fig 3. However, in this latter figure it can be seen that the level of input proteins (both Ets1 and p52) is significantly lower for the R241A assay. Although the quantification in Fig 2B takes this into account this does mean that at face value the R241A experiment is not comparable to the others as the concentration of protein in the IP will be lower. Was this lower concentration of input proteins for the R241A consistent across different repeats? When the authors refer to data being from 3 repeats, were these biological repeats where extracts were prepared and used from separate cell transfections? Or were they technical replicates of separate IPs from the same protein extract? If the latter, then 3 biological repeats need to be performed to generate this data. If the former, is there an example of a co-IP where the levels of p52 and Ets1 in the R241A samples were comparable to the other mutants being tested?

To avoid the problem caused by the different levels of input proteins in IP when the cultured cells were used, we expressed and purified all p52 and ETS1 proteins from *E. coli*. We then used Flag tagged-ETS1₃₃₁₋₄₄₁ to pull-down wild type and mutant p52-RHD. The results are consistent with the previous conclusion. The text (page 9, line 153-156) and figure (Fig. 2b, Supplementary Fig. 3) were revised accordingly.

(2) In Figure 3 the authors conclude that p52 is not contacting a putative p52 binding motif present in the Tert promoter DNA sequence used. I agree with this conclusion. However, the DNA sequence used does not extend in the other direction to where p52 is found in the structure. It is possible therefore that p52 makes contacts with the DNA in this region of the promoter

(even if it does not appear as a putative kB site). It is not clear to me if this sequence was present in the 25 bp Tert promoter oligo used in the ITC experiments in Fig 2C & D. It would, however, be informative if these experiments could be repeated with oligos where the sequences 5' and 3' to the Ets1 binding site are removed to clearly determine if there is a contribution of flanking sequences to the DNA binding affinity of the p52/Ets1 complex.

We thank the reviewer for raising this point. The 25 bp *TERT* promoter oligo used in the previous ITC experiments in Fig 4C & D (not Fig 2c & d) extends 8 bp in the other direction to where p52 is found in the structure, thereby providing DNA sequence for contacting by p52. We followed the reviewer's suggestion and performed ITC experiments with the shorter oligos (listed in Supplementary Table 2) where the 5' and 3' flanking sequences to the Ets1 binding site were removed. The results (Fig. 3b vs Supplementary Fig 8b; Fig. 4c vs Supplementary Fig 8c) showed that the flanking sequences had no contributions to the DNA binding affinity of the p52/Ets1 complex. The text (page 11, line 199-202 and page 13, page 229-233) and figure (Fig. 3b, Fig. 4c and Supplementary Fig. 8b, c) were revised accordingly.

(3) Figures 5D-F present luciferase assay data analyzing the functional effects of p52 and Ets1 on Tert promoter activity. There are a number of concerns about this data.

(a) The figure legend states that these are the average of two independent experiments. This is insufficient and further biological replicates should be performed so that statistical analysis of the data can be performed.

We thank the reviewer for raising an important concern. We have now repeated the luciferase experiments and present the data from three biological replicates in Figures 5d-f. We have also performed statistical analyses of the data.

(b) In Fig 5E and 5F, the additional effects of either expressing NIK or p52 respectively are quite small. This may result from endogenous levels of p52 in the cells, which the authors could assess by western blot combined with p52 siRNA to assess the extent to which the effects of Ets1 are dependent on any endogenous non-canonical pathway signalling. Nonetheless this data would be strengthened if the authors could examine endogenous levels of Tert.

This is an excellent comment from the esteemed reviewer. As suggested, we analyzed the endogenous levels of both p52 and ETS1 in vector control T98G cells and compared their expression levels following NIK or p52 and ETS1 overexpression by western blot, in parallel with luciferase reporter assays in these cells (Fig 5e and 5f). From our western blot analyses (Fig 5e and 5g), we found that moderate levels of p52 are detected while ETS1 protein is highly abundant in these glioblastoma cells. As pointed out by the reviewer, the relatively small effect of expressing NIK or p52, in combination with ETS1 in Fig 5e and 5f may be due to the endogenous levels of p52 and ETS1 observed in the cells. Hence, we have performed the reporter assay in control and NIK-expressing cells treated with NF- κ B2 or ETS1 siRNAs to examine the regulation of p52 and ETS1 on -146C>T mutant *TERT* promoter activation during non-canonical NF- κ B signalling. As shown in Fig 5g, p52 siRNA downregulation in NIK expressing cells resulted in the decline of ETS1 expression, along with the concomitant reduction of -146C>T *TERT* promoter activity. Notably, ETS1 siRNA downregulation during activation of non-canonical NF- κ B signalling resulted in the significant drop of 146C>T *TERT* promoter

activity (Fig 5g). These observations demonstrate that activation of the -146C>T mutant *TERT* promoter by p52 is dependent on the presence of ETS1 sites and *TERT* expression is regulated by non-canonical NF-κB signalling.

(c) The strong effect of the RMK mutant in Fig. 5F supports the idea above that endogenous p52 is active in these cells. However, this experiment lacks a western blot control to demonstrate that the p52 RMK mutant is not affecting the levels of transfected Ets1 in these assays.

We appreciate the reviewer's comment and have included a western blot figure to show that similar levels of transfected ETS1 are expressed in p52 WT and RMK mutant cells in Fig.5f.

(d) A control missing from these experiments is the effect of the p52 RMK mutant on activation of an NF-κB reporter plasmid. Do these mutations specifically affect Ets1 dependent transactivation of the mutant *Tert* promoter or do they have more general effects on p52 dependent gene expression.

We appreciate the reviewer's question and have repeated the luciferase assay using an NF-κB2 reporter plasmid. This was the best experimental strategy in the cells used in our study as no p52 specific endogenous genes could be reproducibly assessed in these cell types (unlike in cells derived from lymph nodes and peyers patches) As shown in Fig 5h, p52 RMK mutant activates the NF-κB2 reporter to a similar extent as p52 WT in human cancer cells under investigation. This finding demonstrate that the mutations in p52 RMK specifically affect ETS1 dependent transactivation of the -146C>T mutant *TERT* promoter, without influencing p52-dependent gene expression.

(4) An important experiment missing from this study is ChIP analysis to demonstrate p52 recruitment to the endogenous mutant *Tert* promoter in cells and whether this is affected by the RMK mutant as would be predicted from the authors' data.

We appreciate the reviewer's question. Due to the presence of endogenous wild-type p52 in the -146C>T mutant cells, it is not feasible to assess ectopic p52 RMK mutant recruitment to the endogenous mutant *TERT* promoter without completely eliminating residual wild-type p52 activity at the mutant *TERT* promoter. One suitable experimental strategy would be to generate endogenous p52 RMK mutant -146C>T cells using CRISPR/Cas9 genome editing to perform the ChIP analysis. However, this will be an independent project by itself and requires 6 additional months, requiring a comprehensive set of detailed validation experiments, which would substantially delay the timely publication of our current findings.

Minor concerns

(5) In Fig 2B it should be clarified if the full length p52 and Ets1 proteins are being expressed in this assay.

As suggested by the reviewer, the sequences of p52 and ETS1 used in Fig.2b has been clarified (page 9, line 153-156).

(6) In Fig 2C the amino acids of the p52 RHD protein should be specified (as different ones are used in the manuscript)

We have specified the p52-RHD (residues 35-329) in Fig. 1a and this is consistent throughout the manuscript.

(7) In the last paragraph of the introduction the authors state that this manuscript provides evidence for the first time that NF- κ B transcription factors can regulate gene expression without binding DNA. However, as discussed by the authors later, this is not the first evidence. However, it is the first structural proof of this.

Thanks for pointing this out. As suggested by the esteemed reviewer, we have revised the text and it now reads “While understanding the reactivation of mutant TERT promoter was the main aim of this study, our results also comprehensively document that NF- κ B transcription factors can regulate transcription without specific DNA binding.” (page 6, line 98-101).

(8) In the discussion in the text of Fig, 5E the authors refer to p52 and Ets1 synergistically activating transcription from the Tert promoter. But as discussed above, the effects of p52 are quite small in these assays and certainly not synergistic.

We appreciate the reviewer’s comment and agree that the effects of p52 ectopic expression are quite modest. Indeed it is well known that TERT promoter is only activated 1-2 fold in cells such as stem cells, germ cells and cancer cells of various origins. Levels of TERT are kept low (as shown by Jerry Shay and Thomas Cech) and it is hence physiological that effects on TERT promoter are perceived as weak effects. But this is in line with all the published literature that TERT promoter is a weak promoter as reactivated TERT has both canonical and non-canonical functions and aberrant activation of this promoter will have severe consequences on transformation of cells. We have excluded to this in the revised text. As illustrated in the western blot analyses above, this is due to the endogenous levels of p52 present in the reporter cells and we have further substantiated our observations of the synergistic regulation of p52 and ETS1 on -146C>T mutant *TERT* promoter activation by performing the reporter assay in NIK-expressing cells following siRNA downregulation of p52 or ETS1. As shown in Fig.5g, ETS1 siRNA knockdown during non-canonical NF- κ B signalling resulted in the significant drop of 146C>T *TERT* promoter activity. These observations demonstrate that activation of the -146C>T mutant *TERT* promoter by p52 is dependent on the presence of ETS1.

(9) The manuscript lacks page numbers.

The page numbers have been added.

Reviewer #2 (Remarks to the Author):

The manuscript “Structural basis for reactivation of -146C>T mutant TERT promoter by cooperative binding of p52 and ETS1/2” is based on novel structural studies that show

interaction of ETS1/2 transcription factors at the de novo site created by a C>T single nucleotide mutation at -146 bp position from the ATG start site. Previously, the same group had shown that non-canonical NF- κ B signaling and ETS1/2 cooperatively drive TERT promoter activation with C>T mutation at the -146 bp position from the ATG site.

In this study, experimental evidence has been provided to show that p52 instead of binding to DNA acts as a co-regulator of ETS1 to inhibit auto-inhibition of ETS1 and forms p52/ETS1 heterotetramer that requires native flanking ETS motifs for sustained activation of the mutant promoter with C>T mutation at the -146 bp position. The impact of this novel observation can be more general and beyond TERT activation as this is the first study showing the inhibitory properties of p52 to counteract auto-inhibition of ETS1 with relevance for activation of promoters that lack κ B consensus sites. Interaction between p52 and ETS1 has not been reported previously and as stated in the manuscript could be due to the weak interaction. Authors have also explained the probable reason for interaction of p52-ETS1 heterotetramer specifically with the mutant sequence at -146 bp and not at -124 bp position.

Ever since the discovery of TERT promoter mutations, the observational data have been consistent. The C>T mutations at -124 and -146 bp, mutually exclusive, occur at high frequencies in cancers that arise from tissues with low rates of self-renewal that include melanoma, bladder cancer, glioma, hepatocellular carcinoma and others. In all cancers with the exception of skin, the -124C>T mutation is predominant. In skin cancers, melanoma and non-melanoma, the two mutations -124C>T and -146C>T occur at almost equal frequencies with -146C>T in some studies shown to occur at slightly higher frequency than the -124C>T mutation. While tumors with either of the two TERT promoter mutations have consistently been shown to have increased TERT expression compared to tumors with those mutations, one study on glioma clearly showed that the increase in TERT expression due to -124C>T mutation was statistically significantly higher than the -146C>T mutations. Those observations point to different operative mechanisms for the two TERT promoter mutations despite creation of similar binding motifs for ETS transcription. It is important authors mention these facts in the manuscript in support of their results either in the introduction or in the discussion section.

The manuscript per se in the current form is quite diffuse, requires streamlining and should include the observations made in the previous paragraph.

We sincerely thank the reviewer for the succinct summary of the significance of our work and positive comments. We also thank the reviewer for the detailed depiction of the fact that -124C>T and -146C>T TERT promoter mutations may have different operative mechanisms. We have added these observations in the discussion (page 17, line 308-315). We also have streamlined the manuscript based on reviewers' suggestions.

Authors should provide values for statistical significance for the results from reporter assays and state whether differences in promoter activities for different constructs were statistically significant.

We thank the reviewer for raising an important concern. We have now repeated all the reporter

assay experiments in this revised manuscript and have presented the data from three biological replicates in all the figures. We have also performed statistical analyses of the data as indicated in the figure legends to show whether differences in promoter activities were statistically significant.

Reviewer #3 (Remarks to the Author):

In this paper, Xu et al. describe biochemical and structural studies of a TERT promoter DNA-bound ETS1-p52 transcription factor (TF) complex. This promoter is activated during cancerogenesis by two mutations that create binding sites for the ETS1-p52 TF complex and the authors aim to understand how this complex selectively recognizes the mutant promoter TF binding site and thus contributes to upregulation of TERT expression in cancer.

It has long been known that members of the ETS and NFkB family of TFs engage through evolutionarily conserved physical interactions to regulate inducible gene expression. How such these TFs cooperate to read out composite DNA binding sites is an important problem in the combinatorial control of gene expression. The authors show convincingly that the dimerization domain of p52 interacts with the DNA binding domain (DBD) of ETS1. Surprisingly, only the ETS1 DBD but not that p52, contributes specific DNA contacts on the co-crystallized TERT promoter fragment. The authors perform biochemical experiments and reporter gene assays to confirm the relevance of the interaction for transcriptional regulation of the TERT promoter. These data indicate that p52 prevents ETS1 autoinhibition and allows tetramerization of the complex through a p52-DD homodimerization interface without itself contributing to DNA binding. These data thus indicate that members of the NFkB family of TFs can contribute to the regulation of gene expression even while not engaging in direct DNA contacts.

It is of no doubt that this work will be of considerable interest to specialists in ETS and NFkB signaling. There are however reservations that arise from the fact that only limited fragments of ETS1, p52 and the DNA elements are co-crystallized and that the biochemical analysis is incomplete, as indicated below. While the fragments are sufficient for the interaction between ETS1- p52 and ETS1-DNA, it remains unclear whether a more extensive cooperative interface of the dimer is formed on an extended single-site DNA substrate, or of the tetramer on an extended two-site DNA substrate. The conclusion that the p52-DBD does not contribute to DNA binding seems premature and requires further analysis that should be readily doable with the available material in the authors' laboratory. Thus the study is promising and I would encourage resubmission of a revised manuscript. Upon revision, the authors should tighten up the manuscript, in particular the Discussion section. Nature Comm should do a careful copyediting job if the MS is accepted, as there are grammatical errors that impede effective communication.

We thank the reviewer for the concise summary of the significance of our study and supportive comments on resubmission of our manuscript. We have tightened up the manuscript including the Discussion section as suggested by the reviewer to the best of our abilities. We are very open to Nature communication for further copy editing our text files.

Major comments:

1. The mutagenesis results shown in Fig. 3 only partially explain binding selectivity: Why does the triple mutant (RMK>A) of p52 show residual binding to ETS1? Would this indicate that the interface is more 'pleiotropic' than assumed by the authors and that other NFkB family members (p50, RelA, RelB) interact with ETS1? Can they test this possibility? Bacterial expression constructs for the RHD domains are readily available in the field.

We appreciate the esteemed reviewer for raising this important point. To address this concern, we purified the RHD domains of p50, RelA, and RelB expressed in *E. coli* and checked their interactions with ETS1 by Flag pull-down assays. The results shown in Supplementary Fig. 5 is consistent with the view that the p52/ETS1 interface is pleiotropic, as all the 4 NF-kB family members can interact with ETS1 with p52 RHD domain exhibiting the strongest binding to ETS1. Although the exact mechanism that only the p52 subunit downstream of non-canonical synergize with ETS1/2 to drive -146C>T *TERT* promoter activation remains to be further investigated, the strong binding affinity between p52 and ETS1 may be a contributing factor in this process. The text (page 9-10, line 166-169) and figure (Supplementary Fig.5) have been revised accordingly.

2. Fig. 3: The authors suggest that the p52-DBSD does not engage the DNA binding site because the potential binding is sterically blocked. Could this be due to crystal packing and the preferential orientation of the co-crystallized complex on the 15bp DNA duplex that enabled successful crystallization? While the authors show DNA binding data that demonstrate that the isolated p52-RHD does not support DNA binding on the minimal 25bp DNA substrate, it remains unclear if the p52-RHD contributes to DNA binding through a cooperative effect involving ETS1. The authors need to show DNA affinity measurements of ETS1 in the presence of p52-RHD to support the conclusion that the p52-DBSD does not contribute (even non-specifically) to DNA binding avidity. This can be easily done by a repeat experiment similar to that shown in Fig. 4C, but using the entire p52-RHD. If there is indeed no contribution to DNA binding by the p52 DBD in solution in the presence of ETS1, the physiological relevance of the unusual arrangement in the crystal structure would be more believable.

We thank the reviewer's comments. We have checked the crystal packing thoroughly and confirmed that the non-engagement of p52-DBSD in DNA binding is not due to crystal packing. As suggested by the reviewer, we performed the ITC experiment shown in Fig. 4C (rearranged as Fig. 3b) using the entire p52-RHD. The results shown in Figs 3b & 3c indicate that the p52 DBD does not contribute to DNA binding.

3. p.10, bottom paragraph: It is not clear how counteraction of ETS1 autoinhibition by p52 enables stronger binding of ETS1 to DNA...! Please clarify.

DNA binding of ETS1 is negatively regulated by the autoinhibitory module flanking the ETS1 domain. But how the autoinhibitory module consisting of four helices (HI-1, HI-2, H4 and H5) inhibits DNA binding of ETS1 is not very clear. One proposed model suggests that the packing of the autoinhibitory module against the ETS1 domain favors a conformation of ETS1 that cannot contact DNA optimally, and the affinity of ETS1 for DNA is inversely correlated with the stability of the inhibitory helices HI-1 and HI-2. Our structure suggests that the binding of p52 to ETS1 would displace and destabilize the HI-1 and HI-2 helices in the autoinhibitory module, consequently disrupting their inhibitory function and enabling stronger binding of ETS1 to DNA.

The mechanism by which p52 counteracts autoinhibition of ETS1 resembles that of Runx1 (Shrivastava et al. 2014, *Leukemia*).

4. The authors need to confirm that the p52-ETS1 complex forms a heterotetramer in solution eg. by performing gel filtration/SEC-MALS etc.

We have tried to perform gel filtration/SEC-MALS using p52/ETS1 complex. However, p52 and ETS1 always are separated in two peaks in gel filtration (Supplementary Fig. 9b), regardless of the pH and salt concentrations used in the buffer. This may arise from the weak interaction between p52 and ETS1.

Disulfide crosslinking has been reported to characterize such weak protein complex. We therefore have performed disulfide crosslinking of the p52/ETS1 complex by mutating two interacting residues (one each in p52 and ETS1) to cysteine in the p52/ETS1 interface (details depicted in methods, page 24-25, line 447-458). SDS-PAGE analysis of the cross-linked sample showed that a specific band corresponding to the p52/ETS1 heterodimer could be observed, and reversing the crosslinking using the reducing agent DTT reverts the p52/ETS1 heterodimer to their respective monomers (Supplementary Fig. 9c). Next we used Blue Native Electrophoresis to check whether the crosslinked p52/ETS1 complex can form a heterotetramer. The result (Supplementary Fig. 9d) showed that the band of crosslinked p52/ETS1 (94 kDa) is significantly higher than that of p52 homodimer and is very close to that of the marker protein (purified human DNMT1, residues 646-1600, 108kDa), indicating that the crosslinked p52/ETS1 complex forms a heterotetramer in solution.

5. Fig. 5a: The DNA configuration in the proposed tetramer is unusual and very different from what is proposed for (the more reasonable) arrangement of the GABPB-GABPA tetramer shown in Fig. 5b. DNA is one of the stiffest of known polymers with a persistence length of ~150 bp. The required DNA contortions would be expected to be energetically unfavorable and it is not clear how the proposed arrangement would enable a ETS1-p52 tetramer to bind two ETS motifs. This casts further doubt on the physiological relevance of the proposed tetramer.

We thank this reviewer for the constructive comments on this point and agree that the required DNA contortions for binding the mutant *TERT* promoter with two ETS1 binding sites would be expected to be energetically unfavorable. However, our EMSA result showed that the purified p52/ETS1 heterotetramer is able to bind a 60bp -146C>T mutant *TERT* promoter DNA with two ETS binding site (Please see our reply to point 6, Supplementary Fig. 9d).

6. Considering that the two ETS site are within reasonably close proximity (~50-60bp), it would be important to test binding of the ETS1-p52 tetramer to such two-site substrates, eg. by EMSA analysis. If the tetramer cannot bind to such sites, as discussed on p.14, is it the *TERT* promoter even the physiologically relevant binding site?

We have recovered and purified the p52/ETS1 heterotetramer from Blue Native gel and performed EMSA to test its binding to the -146C>T mutant *TERT* promoter DNA with two ETS

binding site (60 bp). The result show the p52/ETS1 heterotetramer can interact with such DNA sequence (Supplementary Fig. 9d).

7. It is not clear to this reviewer that the authors have sufficiently rigorously demonstrated that p52 in the p52-ETS1 complex does not contact DNA.

We have sufficient structural and ITC data (Fig 3) showing that p52 in the p52-ETS1 complex does not contact DNA but cannot exclude the possibility that p52 in the p52-ETS1 complex contacts DNA non-specifically. As such, we changed all descriptions of “p52 in the p52-ETS1 complex does not contact DNA” into a more precise description “p52 in the p52-ETS1 complex does not contact DNA specifically” throughout the manuscript. Since the main objective of this study is to delineate the mechanism of p52-ETS1 in the context of the mutant TERT promoter reactivation, and since this will be the first structural demonstration of this phenomenon, we hope other researchers will consider field-testing this model further on genome level and study if such synergy occurs on other physiologically relevant sites in the genome.

8. For ITC data analysis: What model was used for fitting of experimental data? What are the error values and binding stoichiometry?

We used one binding site model for fitting the experimental data. The error values and binding stoichiometry have been added.

Minor comments:

1. Please add page and line numbers to MS.

Page and line numbers has been added.

2. p.2, Line 1: Reference is missing supporting the statement that these are the ‘most common genetic alterations in cancer’.

The ‘most common genetic alterations in cancer’ should be ‘the most common noncoding mutations in cancer’ (revised in page 4, line 54-57). The Reference has been added (Reference 13).

3. p.2, Line 4: The mutations do not reactivate the enzyme TERT but ‘expression of’ the enzyme. Please correct.

The text has been corrected (page 4, line 58-61).

4. p.2, Second paragraph: Reference is missing supporting the statement: ‘...’ reactivation of these two mutant promoters’.

The reference has been added (Reference 16).

5. p.5, Second paragraph: Replace ‘44’ with ‘441’

Corrected (page7, line 114).

6. p.5, Second paragraph and Fig.1: Please use consistent naming of subdomains in text and figures: Eg. p52-DBSD (text) and DBD (Figure) for the same segment.

As suggested, the consistent names of subdomains have been used throughout the text and figures.

7. p.9, bottom paragraph:...conformation will ‘inhibit DNA binding’ or ‘reduce DNA binding affinity’ but not ‘inhibit DNA binding affinity’!

As suggested, it has been rewritten as “reduce DNA binding affinity” (page 12, line 215).

8. Table 1: Refinement section: Rfree value should not be in parenthesis as this indicates the ‘values in the highest resolution shell’

Corrected as suggested by the reviewer (Supplementary Table 1).

We would like to sincerely thank all the reviewers as their incisive comments have strengthened the basic tenets of the manuscript and made the conclusions much stronger.

Reviewers' comments:

Reviewer #1 (Remarks to the Author):

The authors have addressed the issues I raised in my review and I have no further concerns.

Reviewer #3 (Remarks to the Author):

The authors' response partially overcomes my previous concerns with this MS. Remaining issues are as follows:

Major comments:

1. The authors now show that other NFkB family members can interact with ETS1/2. The FLAG pulldowns indicate 2-5x fold lower Coomassie staining for other NFkB family members. Thus, while the p52-ETS1 interaction appears strongest, one can not conclude that (line 171 et seq.): '...only the p52 subunit can synergize with ETS1/2'. In absence of more quantitative data, it is not clear that such strong claims are warranted.

As all 4 NF-kB family members can interact with ETS1 in this assay, it would have been interesting to understand how gene duplication and divergence among NFkB and ETS family members contributes to functional specificity enabling TERT promoter regulation by p52/ETS1. As the triple mutant (RMK>A) of p52 shows residual binding to ETS1, it seems that one has missed the opportunity to fully understand how these proteins interact. It is surprising that the authors did not probe this interaction further, eg. by mutating L328 of p52, a residue that is not conserved among RHDs. Would a RMKL328 quadruple mutant abolish ETS1 binding? Thus, unfortunately, despite the present data, it remains unclear why p52 binds strongest to ETS1 and why these REL and ETS family members appear to interact selectively.

2. The authors' data suggest that the p52-DBSD does not engage the DNA binding site. One concern is that the arrangement seen arises due to crystal packing. The authors' reply 'We have checked the crystal packing thoroughly and confirmed that the non-engagement of p52-DBSD in DNA binding is not due to crystal packing', is not convincing. It is not by checking crystal packing that one can demonstrate that the arrangement seen is physiologically relevant. If indeed such a conclusion can be drawn by the analysis of crystal packing, then this claim needs to be backed up, eg. by showing this in a supplementary figure.

3. It would have been more convincing to analyze a DNA binding mutant of p52 and verify the impact on TERT regulation.

Minor comments:

1. Line 202-203: Do they mean ...'TERT promoter in vitro'. After all, the section describes biophysical measurements of binding interactions in vitro.

2. Section Line 210 et seq.: References to original literature are missing: Eg. Line 213: 'Chemical shift

data...' and Line 216:' Both hydrogen exchange...' Which data are referred to?

3. There are still numerous grammatical errors throughout the text. Nature Comm should do a careful copyediting job prior to publication.

Reviewer #3 (Remarks to the Author):

Major comments:

1. The authors now show that other NFkB family members can interact with ETS1/2. The FLAG pulldowns indicate 2-5x fold lower Coomassie staining for other NFkB family members. Thus, while the p52-ETS1 interaction appears strongest, one can not conclude that (line 171 et seq.): ‘....only the p52 subunit can synergize with ETS1/2’. In absence of more quantitative data, it is not clear that such strong claims are warranted.

As all 4 NF- κ B family members can interact with ETS1 in this assay, it would have been interesting to understand how gene duplication and divergence among NFkB and ETS family members contributes to functional specificity enabling TERT promoter regulation by p52/ETS1. As the triple mutant (RMK>A) of p52 shows residual binding to ETS1, it seems that one has missed the opportunity to fully understand how these proteins interact. It is surprising that the authors did not probe this interaction further, eg. by mutating L328 of p52, a residue that is not conserved among RHDs. Would a RMKL328 quadruple mutant abolish ETS1 binding? Thus, unfortunately, despite the present data, it remains unclear why p52 binds strongest to ETS1 and why these REL and ETS family members appear to interact selectively.

As suggested by the reviewer, we expressed and purified the p52 RMKL328 quadruple mutant from *E. coli* and then used Flag pull-down assay to check its binding to ETS1. The results enclosed below show that p52 RMKL328 quadruple mutant still retains residual binding to ETS1, indicating that quadruple mutations could not abolish the binding of p52 to ETS1. As such, we have toned down the conclusion (line 171 et seq.) by changing ‘....only the p52 subunit can synergize with ETS1/2’ to ‘the p52 subunit downstream of non-canonical NF- κ B signalling appears to be preferentially selected for synergizing with ETS1/2 to drive -146C>T mutant TERT promoter activation’.

Interactions of ETS1 with p52 and its mutants. Flag-ETS1331-441 was used to pull-down p52-RHD and its mutants. The bound proteins were analyzed using SDS-PAGE and Coomassie blue staining. The assays were quantified by band densitometry. Error bars, s.d. for triplicate experiments.

2. The authors' data suggest that the p52-DBSD does not engage the DNA binding site. One concern is that the arrangement seen arises be due to crystal packing. The authors' reply 'We have checked the crystal packing thoroughly and confirmed that the non-engagement of p52-DBSD in DNA binding is not due to crystal packing', is not convincing. It is not by checking crystal packing that one can demonstrate that the arrangement seen is physiologically relevant. If indeed such a conclusion can be drawn by the analysis of crystal packing, than this claim needs to be backed up, eg. by showing this in a supplementary figure.

Although the electron densities of p52-DBSD are not clear enough for unambiguous model building, we can roughly fit the p52-DBSD into these fragmented electron densities (supplementary Figure 2A) and then analyze crystal packing. The results (supplementary Figure 2B) show that p52-DBSD can be comfortably accommodated without steric clashes with p52-DSD, ETS1 and DNA. As p52 has been shown to specifically bind to consensus κ B sites, the non-engagement of p52-DBSD in DNA binding is likely due to the lack of the κ B site in the mutant TERT DNA used for crystallization rather than due to crystal packing.

3. It would have been more convincing to analyze a DNA binding mutant of p52 and verify the impact on TERT regulation.

p52 has been shown to specifically bind to consensus κ B sites and all the current DNA binding mutants of p52 are based on its interaction with consensus κ B sites. As -146C>T TERT promoter does not contain any consensus κ B site and we cannot see the DNA binding of p52 in our structure, it would be difficult, if not impossible, for us to analyze a DNA binding mutant of p52 and verify the impact on TERT regulation. Moreover, performing such an experiment is beyond the scope of this manuscript.

Minor comments:

1.Line 202-203: Do they mean ...'TERT promoter *in vitro*'. After all, the section describes biophysical measurements of binding interactions *in vitro*.

Thanks the reviewer for pointing out this error. We have change 'TERT promoter *in vivo*' to 'TERT promoter *in vitro*'.

2.Section Line 210 et seq.: References to original literature are missing: Eg. Line 213: 'Chemical shift data...' and Line 216:' Both hydrogen exchange...' Which data are referred to?

Both of these two data are referred to Reference 34 and we have added it to the text.

3.There are still numerous grammatical errors throughout the text. Nature Comm should do a careful copyediting job prior to publication.

We have further checked grammatical errors throughout the text as suggested by the reviewer.